# Position: Invisible Tokens, Visible Bills:
# The Urgent Need to Audit Hidden Operations in Opaque LLM Services

Guoheng Sun [* 1]   Ziyao Wang [* 1]   Xuandong Zhao [2]   Bowei Tian [1]
Zheyu Shen [1]   Yexiao He [1]   Jinming Xing [3]   Ang Li [1]

## Abstract

Modern large language model (LLM) services increasingly rely on complex, often abstract operations, such as multi-step reasoning and multi-agent collaboration, to generate high-quality outputs. While users are billed based on token consumption and API usage, these internal steps are typically not visible. We refer to such systems as Commercial Opaque LLM Services (COLS). This position paper highlights emerging accountability challenges in COLS: users are billed for operations they cannot observe, verify, or contest. We formalize two key risks: *quantity inflation*, where token and call counts may be artificially inflated, and *quality downgrade*, where providers might quietly substitute lower-cost models or tools. Addressing these risks requires a diverse set of auditing strategies, including commitment-based, predictive, behavioral, and signature-based methods. We further explore the potential of complementary mechanisms such as watermarking and trusted execution environments to enhance verifiability without compromising provider confidentiality. We also propose a modular three-layer auditing framework for COLS and users that enables trustworthy verification across execution, secure logging, and user-facing auditability without exposing proprietary internals. Our aim is to encourage further research and policy development toward transparency, auditability, and accountability in commercial LLM services.

## 1. Introduction

Large language models (LLMs) have advanced rapidly in recent years, demonstrating strong capabilities in long context understanding (Pawar et al., 2024), reasoning (Chen et al., 2025; Muennighoff et al., 2025), reflection (Renze & Guven, 2024), tool use (Huang et al., 2024; Lu et al., 2024), and planning (Song et al., 2023; Wei et al., 2025). These abilities now enable LLMs to perform increasingly complex tasks (Yuan et al., 2024), often through reasoning and collaboration strategies that resemble human problem-solving (Tran et al., 2025; Feng et al., 2024). Therefore, service pipelines built on LLMs have grown correspondingly sophisticated. Contemporary systems frequently orchestrate extended reasoning chains and coordinate multiple LLM agents to enhance output quality (Tran et al., 2025). However, these intermediate steps are invisible to users, who are billed solely based on token and API usage. We term such invisible computations as *hidden operations*, and define any LLM service that hides its internal steps and returns only the final output as a *Commercial Opaque LLM Service* (COLS). See Figure 1 for an overview of a typical COLS.

COLS conceal their intermediate tokens for three well-established reasons. First, reasoning traces and agentic collaborations are typically verbose and noisy, often containing backtracking, speculative branches, and occasional hallucinations (Zhang et al., 2024; Sun et al., 2025b; Sui et al., 2025). Exposing such raw information could detract from usability or overwhelm users. Second, these traces encode COLS's internal reasoning strategies, tool-using protocols, and multi-agent workflows. Making them public risks model stealing (Carlini et al., 2024; Panda et al., 2024), workflow extraction (Yu et al., 2025; Li et al., 2025), and jailbreaking attacks (Russinovich et al., 2024; Wang et al., 2025a; Xu et al., 2024), compromising the system's intellectual property (IP). Third, abstracting internals allows developers to update backend models, prompts, or tools without changing the user-facing interface, ensuring better scalability.

While these design choices offer clear engineering and user experience benefits, they also introduce systemic challenges for transparency and accountability. Users are charged based

---
[*]Equal contribution [1]University of Maryland, College Park [2]University of California, Berkeley [3]North Carolina State University. Correspondence to: Ang Li <angliece@umd.edu>.

*Proceedings of the 43rd International Conference on Machine Learning*, Seoul, South Korea. PMLR 306, 2026. Copyright 2026 by the author(s).

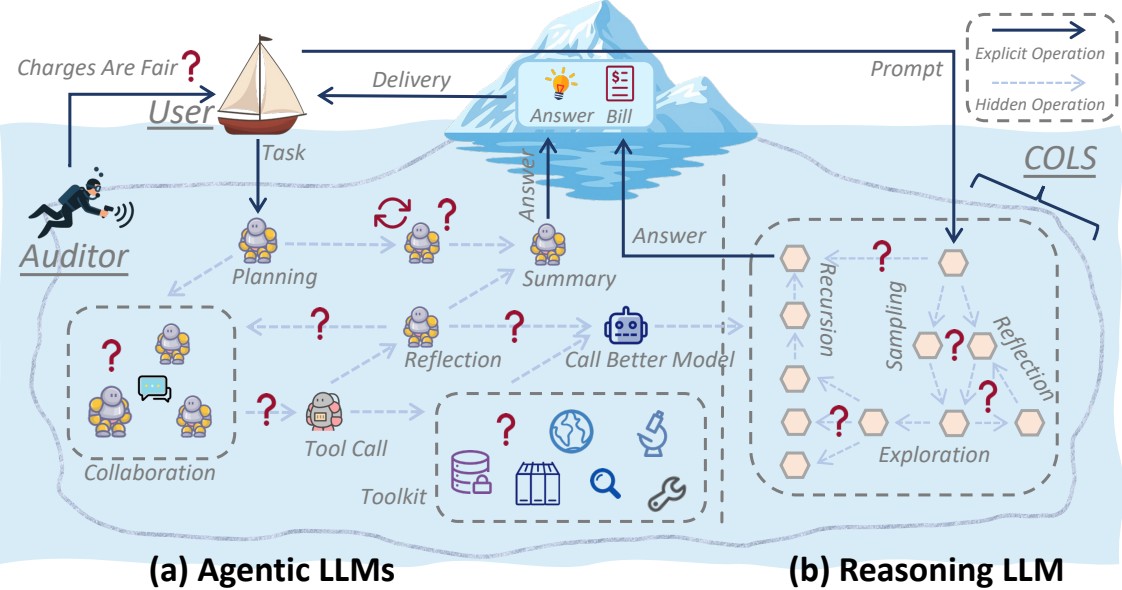

**(a) Agentic LLMs**  **(b) Reasoning LLM**

*Figure 1.* Overview of Commercial Opaque LLM Services and their hidden operations. Part of the illustration was generated by GPT-4o (Hurst et al., 2024).

on the **quantity** and **quality** of operations entirely managed by the service provider, whose incentives are profit-driven. Because these operations are unobservable and unverifiable from the user's side, billing becomes effectively non-auditable and unregulated. In the absence of technical or legal standards, current systems require users to place implicit trust in providers—highlighting the need for verifiable accountability mechanisms. In a competitive landscape where major AI companies increasingly prioritize reasoning and agentic capabilities as profit drivers, this lack of verifiability and regulatory oversight is a serious concern. There exists a fundamental asymmetry between providers and users: users bear financial responsibility for operations they cannot observe, verify, or dispute. Therefore, we make the central claim of this paper: **there is an urgent need to design an auditing framework for hidden operations in COLS.**

In this paper, we examine the *quantity* and *quality* of hidden operations, both of which directly impact billing in COLS. On the quantity side, we identify three forms of potential inflation used by COLS to increase charges: *token count inflation*, *API call inflation*, and *model call inflation*. To detect such manipulations from the user's side, we introduce the concepts of *token auditing* and *call auditing*. On the quality side, COLS may reduce service fidelity to lower internal costs and increase profit. We define two forms of service degradation: model downgrade (Cai et al., 2025) and tool downgrade, and propose model auditing and tool auditing to verify service quality.

For each potential *attack* performed by COLS, we articulate underlying incentives, operational context, and potential harms in the domain of reasoning and agent APIs. For each user-side *auditing* or *defense* strategy, we analyze the challenges across different API settings and propose feasible approaches. Finally, we present a forward-looking research agenda aimed at building a trustworthy framework that balances service quality with user interests. We hope the position taken in this paper can help guide the development of billing verification protocols, regulatory standards, and governance policies for the rapidly growing commercial LLM ecosystems.

Our key contributions are summarized as follows:

- We formalize three billing-related inflation vectors, including token count, API call, and model call inflation, and introduce concrete user-side auditing methods to detect each.
- We identify two forms of service quality degradation, *i.e.,* model and tool substitution, and design model and tool auditing techniques to verify service integrity.
- We analyze the motivations, scenarios, and potential harms of COLS-side manipulations in reasoning and agentic APIs, and map them to corresponding user defense strategies.
- We present a roadmap for building trustworthy LLM services, including technical protocols and policy recommendations for verifiable and transparent billing.

## 2. Background and Problem Formulation

### 2.1. Commercial Opaque LLM Service

COLS are LLM–based services that expose only the final outputs to users while abstracting the underlying computational steps. Such services are typically accessed through

*Table 1.* Visibility and pricing of *reasoning LLM API*'s reasoning tokens. MTok = Million tokens

| Provider | Visible? | Pricing |
|---|---|---|
| OpenAI o1 (Jaech et al., 2024) | ✗ | $60 / MTok |
| OpenAI o3 (Jaech et al., 2024) | ✗ | $40 / MTok |
| OpenAI o1-pro (Jaech et al., 2024) | ✗ | $600 / MTok |
| Gemini 2.5 Pro (Anil et al., 2023) | ✗ | $15 / MTok |
| Claude Opus 4 (Anthropic, 2025) | ✗ | $75 / MTok |

cloud APIs: users submit prompts or tasks to a single end-point and receive a final output, without visibility into intermediate reasoning or operations.

There are two common forms of COLS in practice. The first is the *reasoning LLM APIs*, which encapsulates models designed for complex tasks requiring multi-step inference. These services typically employ models that are optimized with reinforcement learning to improve reasoning depth and answer quality, particularly on complex tasks such as mathematical problem solving and code generation. Although the model may internally perform multiple function calls, speculative reasoning paths, and self-reflections, only the final output is shown to users. Importantly, users are billed based on the total number of tokens generated, including both the visible answer tokens and the unexposed reasoning tokens. As Table 1 shows, some major reasoning model providers charge users for these hidden tokens, based on information available as of May 2025. Although they provide brief summaries generated from the hidden tokens, users remain unaware of the actual reasoning process. Nevertheless, Claude Opus 4 (Anthropic, 2025) encrypts the full reasoning and returns it as a signature, which is a significant advancement and signals a future trend. Our empirical results, summarized in Table 2, show that in current reasoning LLM APIs, the number of hidden reasoning tokens often exceeds the number of answer tokens by more than an order of magnitude. In many cases, more than 90% of the tokens billed to the user are never exposed. This highlights a significant transparency gap and raises questions of billing clarity and fairness.

The second form is the *agentic LLM API*, which enables collaboration among multiple specialized LLM agents. We note that reasoning LLM APIs and agentic LLM APIs are not mutually exclusive categories: an agentic system may internally invoke reasoning models, in which case both forms of opacity coexist, and the audit surface becomes layered, requiring verification of both hidden reasoning tokens and system-level orchestration such as calls, communication, and tool usage. These systems coordinate

*Table 2.* Ratio of reasoning tokens to answer tokens across OpenAI's APIs.

| Model | R/A Ratio |
|---|---|
| o1 | 38.71 |
| o3 | 25.35 |
| o3-mini | 46.33 |
| o4-mini | 25.03 |

agents to solve complex tasks through planning, task decomposition, execution, and summarization. Compared to reasoning LLM APIs, agentic APIs involve more intricate hidden operations. Beyond internal reasoning, agents communicate by exchanging prompts, summaries, and planning instructions. Each agent both interprets inputs from others and generates outputs to guide the workflow. These inter-agent messages may consume substantial tokens, which are often not directly visible to end users. All tokens consumed during agent coordination, including generated prompts, responses, and tool-related instructions, are typically not surfaced to the user. When the agents themselves use reasoning models, billing becomes even more opaque. Moreover, such systems can dynamically substitute or reconfigure tools to reduce backend costs, while continuing to charge premium rates. These behaviors are difficult to detect and audit. These manipulations are difficult to detect, making effective auditing especially challenging in agentic APIs. Table 3 summarizes the pricing models and billing structures adopted by several AI agent providers, with the data reflecting the state of these services up to May 2025. Subscription fees are often tied to credit-based systems, which in turn constrain the number and complexity of tasks that can be executed. However, users are rarely able to determine the true cost of individual tasks.

The most straightforward way to address the auditing challenge is for COLS to directly expose all hidden operations to users. In principle, such full transparency would eliminate ambiguity in both billing and service quality. However, full disclosure of hidden operations is impractical in commercial settings, especially in agentic systems, due to their volume, complexity, and the risk of exposing proprietary models and strategies. As a result, this paper adopts a key assumption: **COLS will not fully expose their hidden operations, or if they do, such disclosure must be protected by mechanisms that prevent extraction, misuse, or unauthorized imitation.** All auditing approaches proposed in this work operate under this practical constraint.

In summary, COLS represent a class of LLM services that prioritize usability, abstraction, and IP protection by hiding internal operations. While this design improves product polish and shields business logic, it also introduces concerns regarding transparency, accountability, and fairness, especially when users are charged for every hidden operation they cannot observe or validate.

### 2.2. Threat Model

In our scenario, we model COLS as potentially misaligned with user interests, not out of malice, but due to profit-driven incentives and structural opacity. COLS may increase the quantity of billed operations or reduce their effective quality, or both, in order to lower operational costs while maintain-

*Table 3.* Pricing plans and billing details of various AI agent providers.

| Provider | Pricing Plan | Pricing Details |
|---|---|---|
| Manus (Manus AI, 2025) | Subscription | $19 / month for 1900 credits, sufficient for completing two to three complex tasks. 300 credits refreshed daily. |
| Relevance AI (Relevance AI, 2025) | Subscription | $19 / month for 10,000 credits. Tasks can use official or custom API keys. Supports customization, but remains hard to audit due to the coarse-grained reporting of LLM APIs. |
| AgentGPT (AgentGPT, 2025) | Subscription | $40 / month. Includes 30 agents per day and 25 loops per agent. |
| Firecrawl's Deep Research API (Firecrawl, 2025) | Pay-as-you-go | $9 for 1,000 credits. Billing is based on number of URLs analyzed 1 credit per URL. |

ing or increasing user charges. The user, as the recipient of the service, and the auditor, as an independent verifier, jointly aim to detect and mitigate such manipulations. Together, they verify the accuracy of the reported quantity of hidden operations and assess the actual quality of service delivered by the COLS. Specifically, given a series of hidden operations triggered by a user request to a COLS, we define the actual quantity of tokens and calls as $T_Q$ and $C_Q$, respectively. Let $T_q$ and $C_q$ denote the unit quality scores of the tokens (determined by the LLMs used) and tools. Then, the fair charge of the COLS, excluding profit, should be $T_Q \cdot T_q + C_Q \cdot C_q$.

The quantities reported by the COLS to the user are denoted as $\hat{T}_Q$ and $\hat{C}_Q$, while the actual service quality (which may be degraded) is denoted as $\check{T}_q$ and $\check{C}_q$. The real cost incurred by the COLS becomes $T_Q \cdot \check{T}_q + C_Q \cdot \check{C}_q$, while the user is charged based on the reported quantities and nominal quality values as $\hat{T}_Q \cdot T_q + \hat{C}_Q \cdot C_q$. By inflating the quantities and downgrading the actual service quality, *i.e.*, $\hat{T}_Q > T_Q$, $\hat{C}_Q > C_Q$, and $\check{T}_q < T_q$, $\check{C}_q < C_q$, the COLS can gain extra profit $P$:

$$P = (\hat{T}_Q \cdot T_q + \hat{C}_Q \cdot C_q) - (T_Q \cdot \check{T}_q + C_Q \cdot \check{C}_q). \quad (1)$$

The user's goal is to audit whether the reported quantities match the actual ones, i.e., $\hat{T}_Q = T_Q$, $\hat{C}_Q = C_Q$, and whether the actual service quality matches the nominal values, i.e., $\check{T}_q = T_q$, $\check{C}_q = C_q$. In this setup, the COLS has access to the user request, the full LLM generation and agent collaboration process, the actual quantity and quality values $(T_Q, T_q, C_Q, C_q)$, and the reported quantity and quality values $(\hat{T}_Q, \check{T}_q, \hat{C}_Q, \check{C}_q)$. In contrast, the user only observes the request, the final output, and the reported values $(\hat{T}_Q, \check{T}_q, \hat{C}_Q, \check{C}_q)$.

### 2.3. Auditing Principles

We suggest a reoriented design philosophy for auditing COLS, one that views auditing as a core capability of system design. There are several general principles for the auditing process:

- **COLS IP Preservation.** To protect the provider's interests, the auditing process should safeguard the confiden-

tiality of internal operations, including reasoning traces, agent workflows, and proprietary toolchains that may be sensitive to reverse engineering or IP concerns.

- **Service-Integrated Verifiability.** Auditing should be seamlessly embedded into the user experience. The system should not only certify billing correctness but also provide users with interpretable confidence metrics, enabling informed trust without accessing internal details.

- **Low False Positive Rate.** Auditing methods should minimize unwarranted flags. Incorrectly flagging honest service providers as misreporting can undermine trust in the auditing framework and create unnecessary friction in commercial deployments.

- **Efficiency and Scalability.** Auditing mechanisms must be practically deployable at scale. They should introduce minimal latency or cost overhead, and remain adaptable across diverse LLM service architectures and usage models.

These principles reflect a normative position: that as LLM services grow in complexity and economic significance, verifiability and transparency should be embedded into their governance and system design.

## 3. Quantity Inflation and Auditing of Hidden Operations

In this section, we define the possible inflation behaviors related to the quantity of hidden operations in COLS, which may result in $\hat{T}_Q > T_Q$ or $\hat{C}_Q > C_Q$ in Eq. 1. We focus on two key forms of inflation: **token inflation** and **call inflation**, and analyze how they may manifest in reasoning LLM APIs and agentic LLM APIs. We then identify the core challenges in auditing these quantities from the user's perspective. Finally, we discuss potential solutions for detecting and mitigating such inflation through targeted auditing strategies.

### 3.1. Reasoning LLM: Token Inflation and Token Auditing

We define the behavior that COLS increases the number of hidden tokens to inflate billing without necessarily improving the answer quality as *token inflation*. We identify

two primary forms of token inflation. The first is *naive inflation*, in which the provider simply overreports the token count without changing the underlying content. Recent work has shown that even the tokenization of a given string is not unique, creating a structural incentive for providers to misreport token counts under pay-per-token pricing (Velasco et al., 2025b). The second is *adaptive inflation*, where the provider appends low-effort or irrelevant content to the reasoning trace. These additional tokens may include duplicated steps, off-topic retrieval results, or meaningless filler text, crafted to evade simple statistical checks. This also includes inserting prompt phrases (e.g., "think as many steps as you can") that implicitly induce the model to generate unnecessarily long reasoning traces without injecting any fabricated tokens. This kind of inflation happens even if COLS release the hidden reasoning tokens.

The potential risk of token inflation underscores the urgent need for **token auditing** for COLS. A token auditing mechanism should verify that the number of reasoning tokens reported by COLS corresponds to meaningful internal computation. Given the user prompt, the final answer, and the reported token count, auditing should assess whether the total number of hidden tokens falls within a reasonable range and whether these tokens make substantive contributions to the final output. Such auditing must not rely on access to the full reasoning trace, and must operate under asymmetric information. This calls for new designs that combine model-based estimation, statistical analysis, and content relevance checks, all while preserving provider confidentiality.

### 3.2. Agentic LLMs: Call Inflation and Call Auditing

Agentic LLM APIs coordinate multiple specialized LLM agents to solve complex tasks through multiple LLM calls and tool invocations, most of which are hidden from the user. However, all these internal LLM calls, model-to-model messages, and tool executions contribute to the final billing. This creates new opportunities for unjustified overhead through what we refer to as *call inflation*.

Call inflation in agentic systems can take several forms. The most direct is *model call inflation*, where the provider either makes excessive model calls, for example by splitting reasoning into unnecessarily fine-grained subtasks or repeating subqueries, or overreports the number of such calls without actually executing them. Another form is *communication inflation*, where agents exchange verbose or redundant messages that generate additional token usage. These messages may be genuinely produced or artificially claimed, yet contribute little to actual task completion. A third form is *tool call inflation*, where external tools are invoked excessively or irrelevantly, or where the reported number of tool interactions is inflated to simulate complexity or justify higher billing.

These forms of inflation are difficult to detect, especially since users have no visibility into the internal workflow, agent structure, or the tool interfaces being used. As a result, users may unknowingly pay for inflated agent interactions and unnecessary tool calls that do not improve the quality of the final answer. This motivates the need for **call auditing** mechanisms tailored to agentic APIs. A call auditing framework should allow users to assess whether the number and type of internal calls reported by COLS are justified by the complexity of the input task and the content of the final output. Auditing should also consider whether the communication patterns and tool usage are consistent with efficient task execution, rather than artificially inflated for billing.

As with token auditing, call auditing must operate under asymmetric information, without access to proprietary agent configurations or execution traces. Designing such mechanisms requires new strategies for estimating agent behavior, benchmarking task complexity, and validating reported usage patterns while respecting the confidentiality constraints of commercial services.

### 3.3. Challenges of Quantity Auditing

Auditing the quantity of hidden operations in COLS presents several key challenges:

- **Limited observability.** The internal reasoning traces and agentic workflows are entirely opaque. Auditing must rely solely on observable information, such as the user prompt, final answer, billing metadata, and the declared service identity. This limited visibility may necessitate a trusted auditor with partial access to internal information, such as proxy datasets or encrypted usage records.
- **High variability of LLMs.** LLM services exhibit significant randomness in computation. Even with identical prompts, the number of reasoning tokens or internal calls can vary across runs. This stochasticity makes it difficult to determine a reliable ground truth for expected usage. Auditing methods based solely on input length or task type may result in high false positive rates. For example, our experiments in Table 4 indicate that a regression neural network cannot accurately predict the number of reasoning tokens given only the length of the prompt and the answer, even when trained on a large-scale reasoning dataset.
- **Adaptive inflation.** COLS may inject tokens or calls that appear superficially relevant but provide little actual value to the output. These low-cost, semantically plausible additions are difficult to distinguish from legitimate computation. Detecting such subtle inflation requires sensitive auditing methods capable of capturing fine-grained differences without introducing excessive false alarms.

*Table 4.* Reasoning token length prediction accuracy on multiple datasets from DeepSeek-R1 (DeepSeek-AI, 2025) using two-layer neural networks. Classification predicts discrete length bins (9–12 per dataset), while regression is considered accurate if within 25% error of the ground truth. All accuracies are below 50%. These results serve as a proof-of-difficulty rather than a strong upper bound: they illustrate that predicting hidden reasoning token counts from limited observable information is intrinsically challenging, even with supervised learning on large-scale reasoning data (Section 3.3).

| Tasks | R1-Math (Hugging Face, 2025) | R1-Coding (Open Thoughts Team, 2025) | R1-Medical (Chen et al., 2024) | R1-General (Glaive AI, 2025) |
|---|---|---|---|---|
| Classification | 22.26 | 33.88 | 43.95 | 25.52 |
| Regression | 26.82 | 29.30 | 20.50 | 19.88 |

## 3.4. Possible Solutions

No single method is sufficient to fully address quantity auditing in COLS. Instead, we propose a combination of complementary strategies: *commitment-based auditing* and *predictive auditing*, which approach the problem from opposite sides—one from the COLS's commitments and the other from the user's expectations. Concurrent work has also explored sequential statistical testing to detect token misreporting by providers (Velasco et al., 2025a). A stronger auditing framework should integrate both, and may further benefit from additional techniques such as behavioral auditing, signature-based auditing, and TEE-based auditing in high-stakes settings. In addition to these auditing strategies, a third line of work, *watermarking*, becomes viable when COLS providers (especially those deploying reasoning LLMs) are willing to expose partial or redacted internal token traces. In such scenarios, watermarking techniques (Kirchenbauer et al., 2023; Zhao et al., 2023) can embed lightweight, verifiable signatures into the generated content to enable downstream verification of both authenticity and integrity.

**Commitment-based auditing** has been exemplified by CoIn (Sun et al., 2025a) framework, which commits to hashed fingerprints of hidden reasoning tokens and enables third-party verification via Merkle-tree (Merkle, 1987) proofs. More generally, commitment-based auditing relies on the COLS provider to generate cryptographic commitments to its internal operations. During the inference process, the provider constructs secure summaries of reasoning tokens, model calls, and tool usage. These commitments are exposed to the user or a third-party auditor in encrypted or abstracted form, allowing selective verification of usage claims without revealing the full trace. Such methods preserve confidentiality while enabling provable consistency between reported and actual operations. The commitment-based auditing requires COLS' cooperation and introduces additional infrastructure and protocol complexity. The main limitation of commitment-based auditing lies in its limited ability to detect adaptive inflation. If COLS injects low-cost fabricated tokens or calls during generation, prior to the construction of secure summaries, these operations may still be faithfully committed and thus bypass verification. In such cases, commitment-based auditing may need to be complemented by additional semantic checks to identify operations that appear valid structurally but contribute little to the final output.

**Predictive auditing**, as exemplified by the PALACE (Wang et al., 2025b) framework, allows users to independently estimate hidden reasoning token or call usage directly from prompt–answer pairs without relying on provider-side commitments. This strategy uses learned models or statistical baselines to predict a plausible usage range, then checks whether the reported quantity falls within this range. For example, an LLM may be trained to estimate the expected number of reasoning tokens given the prompt, answer, and the answer correctness, or to predict the typical number of agent calls for tasks of similar complexity. Predictive auditing does not require access to internal traces or provider cooperation, but it may suffer from uncertainty, especially on diverse or highly stochastic tasks. A key limitation of predictive auditing is its reliance on proxy training datasets to estimate reasonable token or call usage. Since users do not have access to internal reasoning traces, they cannot directly supervise the predictive models. To enable meaningful estimation, COLS may need to release representative data samples, including prompts, outputs, and the associated usage statistics. Without such data, predictive auditing may struggle to produce accurate or generalizable estimates, particularly for diverse task types or proprietary model behaviors.

**Watermarking**, in contrast to the above two, is not feasible in fully opaque settings but offers a powerful enhancement when COLS providers are willing to expose *partial* internal traces. In such cases, watermarking techniques provide a lightweight and effective means to embed verifiable signals directly into model outputs or intermediate steps (Kirchenbauer et al., 2023; Zhao et al., 2023). These signals can assist downstream users or auditors in confirming the authenticity and provenance of results, and in detecting unauthorized content injection. Beyond provenance tracking, watermarking also serves as a practical tool for intellectual property protection. Recent studies show that carefully designed watermarks and sampling strategies have the potential to deter unauthorized model distillation (Zhao et al., 2022; Savani et al., 2025; Pan et al., 2025). By making outputs traceable or resistant to distillation, watermarking helps preserve the integrity of high-value models. In sum, watermarking is not a *general-purpose solution* for opaque COLS but becomes a potent auditing and protection mechanism when partial observability is permitted, serving as a

bridge between full transparency and strict confidentiality.

# 4. Quality Downgrade and Auditing of Hidden Operations

The COLS performs quality downgrade by committing to providing the user with a service of quality level $T_q, C_q$ but generate the answer in a lower quality level $\check{T}_q, \check{C}_q$, allowing the provider to profit from the difference in cost. This downgrade is invisible to the user but has significant impact on service fairness, especially when users are billed as if top-tier resources were used. Since the performance level of modern LLMs is difficult to evaluate using limited samples and fixed benchmarks, quality downgrade is even easier for COLS to implement than quantity inflation. In this section, we analyze *model downgrade* in reasoning LLMs and *tool downgrade* in agentic systems. We identify the core challenges in detecting such downgrade and discuss possible solutions.

## 4.1. Reasoning LLM: Model Downgrade and Model Auditing

In reasoning LLM APIs, providers often maintain multiple variants of the same model family, differing in capacity, training data, or optimization strategy (*e.g.,* ChatGPT o1, o3). Model downgrade refers to the silent substitution of lower-cost models, which may introduce misalignment between expected and actual service quality. For example, a prompt may be processed by a smaller-sized model, while billing remains unchanged. This practice is difficult for users to detect, as the final answer may still appear plausible for many tasks. However, over time, such downgrade can lead to subtle reductions in answer correctness and factual accuracy. The lack of output deviation in simple tasks makes downgrade especially dangerous in high-stakes settings where users expect consistent high-quality reasoning.

To address this issue, **model auditing** should evaluate whether the quality of the underlying model used by COLS matches the claimed or expected configuration. Since users cannot access model internals, model auditing must rely on behavioral cues such as reasoning patterns, failure cases, and performance on calibrated challenge prompts. It may also involve response fingerprinting or output signature estimation to match against known model behavior.

## 4.2. Agentic LLMs: Tool Downgrade and Tool Auditing

In agentic LLM systems, tool usage plays a central role in enabling accurate and verifiable problem solving. Tools may include web search, code execution, database lookup, or external APIs. Tool downgrade occurs when the provider substitutes or disables these tools in favor of cheaper or offline alternatives, while still charging the user as if full tool access were provided. In addition to model downgrade, which may happen within individual agents, tool downgrade introduces another dimension of hidden quality degradation. In some cases, COLS may even simulate tool usage by fabricating plausible answers without actually invoking the tool, further reducing cost while maintaining the appearance of tool interaction.

For example, a call to a live calculator API may be replaced with a local approximation module, or a web search may be omitted entirely and replaced with static retrieval. In some cases, the tool call may be simulated in the trace without actually invoking the backend. These modifications can significantly reduce operational cost but also degrade answer quality or freshness, particularly for knowledge-intensive or real-time tasks.

**Tool auditing** aims to verify whether the advertised tools were actually used, and whether the responses reflect genuine tool outputs. Since tool executions are hidden, auditing must infer tool usage based on answer structure, timing signals, and comparison against known tool response patterns. Detecting simulated or skipped tool calls requires robust signatures of real tool interaction that cannot be easily mimicked.

## 4.3. Challenges

Auditing quality downgrade presents several distinct challenges:

- **Lack of reference outputs.** Quality auditing lacks ground truth outputs to compare against. Users often cannot tell whether a different model or tool would have produced a better answer, especially on subjective or open-ended tasks.
- **Behavioral similarity.** Downgraded models and tools can still produce fluent and plausible outputs. The differences between high-quality and downgraded responses may be subtle, task-dependent, or only observable in aggregate over many queries. This makes downgrade hard to detect with single-sample audits.
- **Sampling stochasticity.** LLMs often use stochastic decoding (e.g., temperature, top-k), so the same input can yield different outputs each time. This randomness makes it hard to tell if a lower-quality response is due to true model degradation or just natural variation. It adds noise to audits and complicates fair comparisons.

## 4.4. Possible Solutions

We outline three complementary strategies for auditing quality downgrade: *behavioral auditing*, *signature auditing*, and *TEE-based auditing*.

**Behavioral auditing** seeks to detect downgrade by analyzing specific response patterns. By submitting calibrated prompts, measuring reasoning depth, tracking accuracy on known benchmarks, users can infer whether the underly-

ing model or tool matches the claimed quality. Behavioral auditing may also leverage LLM-based judges to compare responses across services or against known baselines.

**Signature auditing** relies on hidden but detectable artifacts that distinguish models or tools. These may include stylistic fingerprints, output entropy patterns, or timing signals that reveal whether a real tool was used. Providers could optionally embed verifiable usage signatures into responses, which users or auditors could extract and verify without exposing internal details.

**TEE-based auditing** provides a hardware-secure mechanism for verifying model identity or tool usage without exposing internal logic (Cai et al., 2025). By executing parts of the COLS pipeline within Trusted Execution Environments (TEEs), providers can generate attested summaries that external auditors can verify with strong integrity guarantees (Schnabl et al., 2025). Unlike behavioral or signature-based methods, TEE-based approaches offer cryptographic assurance under confidentiality. While modern TEEs introduce minimal overhead (*e.g.,* under 3% throughput loss), they require enclave-enabled infrastructure and standardized attestation protocols. As such, TEE-based auditing is best suited for high-stakes deployments where strong auditability outweighs deployment complexity.

All approaches face challenges in generality and robustness, but together they offer a path toward holding COLS accountable for quality degradation. As commercial LLM services continue to evolve, we argue that auditing quality is just as important as auditing quantity in ensuring fairness and transparency for users.

## 5. Blueprint for Auditing Frameworks

To enable trustworthy and practical auditing of hidden operations in COLS, we propose a three-layer architectural framework that spans the entire lifecycle of COLS interaction, from service execution and secure logging to external verification and user-facing feedback. This framework is designed to support both reasoning LLM APIs and agentic LLM APIs, incorporating the auditing strategies discussed in previous sections.

**Layer 1: COLS Service Execution.** This foundational layer includes all operations performed by the COLS provider in response to a user query, such as token generation, model calls, inter-agent communication, and tool usage. Some operations are opaque to users but determine both functional outcomes and billing. Providers maintain complete control over these execution strategies, which makes independent verification essential.

**Layer 2: Secure Commitment and Recording.** Upon task completion, the COLS, possibly under auditor supervision,

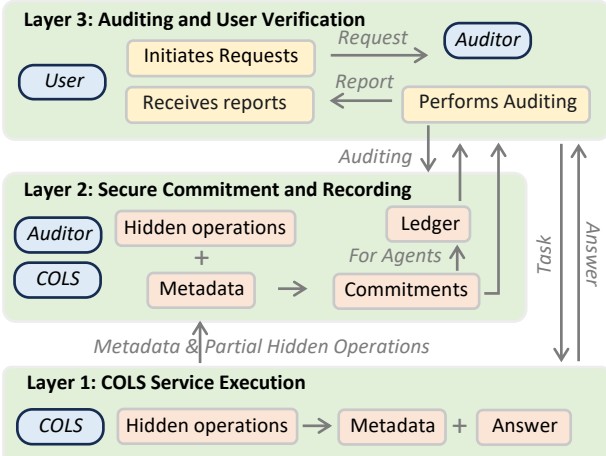

*Figure 2.* Three-layer architecture of the auditing framework. Layer 1 handles execution, Layer 2 generates verifiable commitments, and Layer 3 provides auditing services.

encodes internal operations into verifiable commitments, including hashed reasoning traces, semantic embeddings, or encrypted call logs, following standardized auditable protocols. In agentic settings, each agent's commitments can be anchored into a shared and tamper-resistant ledger using blockchain or similar infrastructure, ensuring traceability across the multi-agent workflow. The commitment process must be transparent and deterministic, preserving confidentiality while enabling verifiability.

**Layer 3: Auditing and User Verification.** The final layer supports external verification and user-facing auditability. An auditor, either a third-party service or part of the user platform, verifies token usage, model identity, or tool behavior based on the commitments produced in Layer 2. Crucially, this layer is *modular*: it supports a wide range of auditing techniques, including commitment-based verification, predictive estimation, behavioral analysis, and signature-based detection, as well as complementary measures such as watermarking and TEEs. New auditing tools can be flexibly integrated into this layer as models and usage patterns evolve. Users interact with the auditor to initiate verification requests and receive audit reports, enabling transparency and dispute resolution without accessing proprietary internals.

## 6. Alternative Views

We acknowledge several alternative perspectives that challenge or qualify the need for technical auditing of hidden operations in COLS.

**Market competition and self-correction.** A natural counterargument is that the competitive AI market already deters systemic billing manipulation: providers have strong long-term incentives to maintain brand reputation and user trust, and those who consistently overcharge will lose customers.

For example, automobile manufacturers and streaming platforms do not disclose production costs, yet market competition effectively disciplines pricing. However, the analogy is imprecise. In those industries, users pay a negotiated price for a finished product or a flat subscription, not per internal production unit. In COLS, users are charged based on *specific internal units*, such as token counts, model calls, and tool invocations, yet have no means to verify whether the reported quantities are accurate. The core issue is not about disclosing production costs, but about enabling users to *verify the billable units* on which they are charged. Combined with high barriers to entry (Vipra & Korinek, 2023) and significant information asymmetry (Erlei et al., 2026), market forces alone may be insufficient to correct subtle misalignment. We view market discipline and auditing as complementary rather than mutually exclusive.

**Auditability versus usability and abstraction.** The primary benefits of COLS, including usability, abstraction, and integrated tooling, may be inherently at odds with auditability. Verification protocols could introduce latency and cost overheads, potentially compromising user experience. Many users prioritize result quality and convenience over billing transparency, willingly accepting opacity as they do with other SaaS platforms (Herrera & Calderón, 2025). This is a legitimate concern. We argue not that every request must undergo full auditing, but that auditability should be *demand-driven and risk-tiered*: lightweight or no auditing for routine interactions, and stronger verification for high-stakes, enterprise, or regulatory-sensitive use cases.

**Outcome-based value and alternative billing models.** A related view holds that users fundamentally purchase *outcomes*, not computation. Much like legal or consulting services, what matters is whether the task is completed satisfactorily, not how many internal steps were taken. From this perspective, alternative billing models such as per-query or subscription-based pricing could improve transparency by shifting pricing responsibility to the provider side, reducing the relevance of token-level auditing. We agree that outcome-based pricing mitigates some concerns. However, it does not fully eliminate the need for auditing: task complexity varies significantly across queries, quality downgrade remains possible regardless of the pricing model, and in practice most commercial services adopt hybrid billing structures that still tie charges to internal resource consumption (Demirer et al., 2025). Whenever hidden operations directly affect billing or service quality, the case for auditability persists.

**Auditing may introduce new trust and privacy risks.** Any auditing framework introduces an additional trusted party: the auditor. Commitment logs, usage metadata, and partial execution traces, even when encrypted or abstracted, may reveal sensitive information about user queries, provider

workflows, or tool outputs. In agentic settings, where systems interact with databases, web services, and enterprise tools, audit records could be more sensitive than the final answers themselves. This is a valid concern. Auditing mechanisms must be designed with *minimal disclosure* principles: auditors should receive only the information strictly necessary for verification, and commitments should prevent reconstruction of proprietary logic or user data. Techniques such as encrypted logs with selective disclosure and strict access control can help mitigate these risks. The goal is not to eliminate trust assumptions, but to distribute them more transparently than in the current model.

**Institutional mechanisms may be more practical than technical auditing.** Finally, one may argue that contractual service-level agreements, third-party certification, periodic compliance audits, and regulatory reporting offer a more scalable path to accountability than cryptographic commitments or behavioral detection. This governance-first perspective draws on established practices in financial auditing and cloud service compliance (Brundage et al., 2026). We view institutional and technical approaches as complementary rather than competing: technical auditing mechanisms can provide the *verifiable evidence* that institutional frameworks require, while institutional frameworks supply the legal and organizational context in which technical tools operate.

# 7. Conclusion

As LLM services grow in sophistication and economic importance, risks from opaque, unverifiable internal operations also increase. Current COLS often obscure internal reasoning and decision-making, limiting users' ability to assess the quantity and quality of service. In this position paper, we highlighted two key risks, quantity inflation and quality downgrade, and proposed auditing strategies grounded in realistic threat models and technical constraints. We presented a taxonomy of mechanisms that balance provider confidentiality with user verifiability, and introduced a three-layer auditing framework for verifiable yet privacy-preserving commitments to internal actions.

We encourage the research community to recognize COLS auditability as a foundational challenge. Future LLM services must incorporate secure commitments, verifiable summaries, and user-accessible audit interfaces as integral parts of their infrastructure. Such architectural changes can play a crucial role in promoting fairness, transparency, and trust in the next generation of intelligent systems.

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
