# OpenReview forum: "Position: Invisible Tokens, Visible Bills: The Urgent Need to Audit Hidden Operations in Opaque LLM Services"
_ICML.cc/2026/Position_Paper_Track — ICML 2026 Position Paper Track regular_

### Official Review · Reviewer_7r5y · 2026-03-14

**Significance:** 4
**Argument Clarity:** 4
**Rating:** 5
**Confidence:** 4

**Questions:**

What's the current research progress on agentic LLMs' tool downgrade, call inflation, and auditing problems?

**Alternative Views Section:**

Yes

**Compliance With Llm Reviewing Policy A Conservative:**

Affirmed.

**Discussion Potential:**

3

**Final Justification:**

The paper has valuable contribution and the authors have fully addressed my concerns. I will maintain the positive score.

**Paper Summary:**

This paper proposes that Commercial Opaque LLM Services (COLS) creates a key challenge: users are billed for opera-tions they cannot observe, verify, or contest, and identifies two risks: quantity inffation and quality downgrade. To address the challenge, the authors suggest a diverse set of auditing strategies, including commitment-based auditing, predictive auditing, and watermarking. Finally, the authors propose a three-layer architectural framework for trustworthy and practical auditin in COLS.

**Position:**

Yes

**Position In Title:**

Yes

**Related Work:**

3

**Strengths And Weaknesses:**

Strengths:

- The paper addresses a timely, important and under-explored topic: opaque LLM services should be auditable. The topic is particularly important especially under agentic AI scenario.
- The paper is well-written and easy to follow. It is good to structure the proposition from quantity inflation and quality downgrade. The problem formulation, threat model, and auditing principles are clearly defined.

Weakness:

- The two-layer neural networks baseline is a bit weak to support the challenge of quantity auditing.
- There seems to be few literature references for agentic LLMs' tool downgrade, call inflation, and auditing problems.

**Support:**

3

---

> ### Author Rebuttal · Authors · 2026-03-31
>
> Thank you very much for the reviewer’s positive evaluation and for recognizing the importance and timeliness of our work.
>
> > # 1. Weak baseline for quantity auditing
>
> We agree with the reviewer that the current two-layer neural network baseline is relatively simple. Our goal in including this experiment was **not** to present a strong final solution or to claim near-optimal prediction performance. Instead, we used it as an initial empirical example to show that **predicting the number of hidden reasoning tokens from limited observable information is intrinsically difficult**.
>
> In other words, this experiment is mainly meant to support the claim that **quantity auditing is hard**, rather than to build a highly competitive predictor. We agree that this point can be stated more clearly in the paper. In the revision, we will clarify that this baseline should be viewed as a **proof-of-difficulty**, not as a strong upper bound.
>
> > # 2. Limited related work on tool downgrade, call inflation, and auditing in agentic LLMs
>
> We thank the reviewer for raising this point. We agree that the current version can better organize the related work on agentic systems. Our goal is to discuss the auditing problem of **Commercial Opaque LLM Services** from a broader perspective, covering both reasoning APIs and agentic APIs, but we agree that the agentic side can be strengthened.
>
> In the current draft, we already cite several closely related directions, including:
>
> - work on **tool-use benchmarks** and **tool-utilization capability**, which helps describe what correct and sufficient tool use should look like;
> - surveys on **multi-agent collaboration mechanisms**, which help explain why agentic systems introduce more hidden operations and more complex billing surfaces;
> - work on **trustworthy LLM agents**, **agent attacks**, and **workflow extraction**, which supports our argument that hidden agent workflows raise not only economic concerns but also security and trustworthiness concerns.
>
> In the revision, we will expand this discussion and make the connection to **tool downgrade**, **call inflation**, and **auditing** more explicit.
>
> > # 3. Current research progress on tool downgrade, call inflation, and auditing
>
> Our short answer is that **this area is still at an early stage**, and progress is quite uneven across subproblems.
>
> - The **more developed** line of work focuses on auditing hidden reasoning or model substitution in relatively controlled API settings.
> - The **emerging** line of work studies predictive or commitment-based estimation of hidden usage.
> - The **still largely open** problems include agent-specific auditing issues such as **tool downgrade**, **simulated tool use**, **redundant multi-agent communication**, and **inflated call structures** in black-box commercial systems.
>
> This is also why we position the paper as a **position paper**, rather than a paper claiming that these problems have already been fully solved. Our goal is to clearly define the threat model, unify the attack surface, and provide a roadmap for future research before these opaque billing and service-integrity issues become more deeply embedded in commercial agent ecosystems. We will make this assessment of the current research progress clearer in the revision.

---

> > ### Author Rebuttal · Reviewer_7r5y · 2026-04-01
> >
> > The authors have address my concerns, and I will maintain my positive scores.

---

### Official Review · Reviewer_P6bi · 2026-03-15

**Significance:** 1
**Argument Clarity:** 2
**Rating:** 5
**Confidence:** 4

**Questions:**

* Why not just bill per query? This would increase transparency and put the onus on the COLS.
* How does low, medium, and high reasoning relate?
* We don't know the parameter counts, number of layers, etc. of these frontier models. This doesn't increase in those directions?
* What can a user do with the auditing information? Use a different model? Why not just check pricing and solution quality of different models?
* Why is this the right question to ask? There are so many facets of COLS that lack transparency. Why is the number of tokens and calls the right number, especially when they give you the price/token and bill you the price...?

**Alternative Views Section:**

Yes

**Compliance With Llm Reviewing Policy A Conservative:**

Affirmed.

**Discussion Potential:**

2

**Final Justification:**

The rebuttal addressed my concerns and I do believe that the main criteria for a position paper has been met: "present a compelling position that warrants greater exposure within the machine learning community".

**Paper Summary:**

This paper argues that closed-source models provided by frontier labs (COLS) lack transparency in pricing and hide the number of tokens used by the model during reasoning and agentic tasks. The authors argue that there should be transparency methods for auditing potential token inflation charged by frontier labs, since they have an economic incentive to inflate the number of tokens being used. The paper sketches potential reasonable solutions for these approaches. Finally, the authors lay out a framework for auditing the hidden operations of COLS and how they should be implemented.

**Position:**

Yes

**Position In Title:**

Yes

**Related Work:**

1

**Strengths And Weaknesses:**

#### Strengths
* The paper is extremely well-written
* The paper has good intentions of increasing transparency and auditability of closed-source models

#### Weaknesses
* It's unclear how one would use the auditing information to make decisions.
* If price is a proxy for tokens used for reasoning and tool-calling, why doesn't a user just use the cost instead of auditing the model.
* The alternative views section is extremely impoverished. There are several other alternative views including policy stating that providers charge by query instead of by number of tokens. Another could be that this is what one has to accept if they want to work with these model providers.
* It's unclear if this paper is at all relevant to the ICML community. This paper is mostly about transparency of financials.
* The COLS are not forcing you to do anything, why do they need to have auditing information about the number of tokens they are using?
* There's no related work section.

**Support:**

2

---

> ### Author Rebuttal · Authors · 2026-03-31
>
> Thank you very much for the careful reading of our paper and for the detailed and constructive feedback.
>
> > # 1. How auditing information helps decisions
>
> We agree this should be clearer. Our main point is that **when billed internal operations are hidden, price alone is not enough for informed decisions**. Auditing helps fill this gap.
>
> It can support: **procurement** (compare not only price/output, but also billing credibility), **dispute handling** (question unusually high hidden usage), **long-term monitoring** (detect gradual increases in billed hidden operations or silent model/tool downgrade), and **governance/compliance** (require evidence that a claimed model/tool was actually used). We will state this more clearly in the revision.
>
> > # 2. Why not just look at cost?
>
> Our core point is: **price is what the provider reports; auditing asks whether that result is verifiable and fair**.
>
> The same price may correspond to different cases:
> - the provider truly used the claimed hidden reasoning/tool usage;
> - the provider over-reported hidden tokens/calls;
> - the provider used a cheaper model/tool but charged the same.
>
> These cases are hard to distinguish from the bill alone. So the issue is not whether users can see the price, but **whether the price can be verified as reasonable**. We will make this sharper in the revision.
>
> > # 3. Why not bill per query?
>
> We agree this is an important alternative view and will add it more clearly. However, **per-query billing does not fully remove the auditing problem**:
>
> - task complexity differs greatly across queries;
> - quality downgrade can still happen;
> - real products often use mixed pricing (subscription, credits, per-tool, token-based).
>
> So we are **not** arguing token-based billing is the only good model. Our point is that **whenever hidden operations affect price or service quality, auditability matters**.
>
> > # 4. Alternative views are too weak
>
> We agree. We will expand this section substantially. In particular, we will add views that:
> - per-query or subscription billing may be better in some markets;
> - some users may accept opacity as a trade-off for convenience and integration;
> - competition and provider reputation may already limit systematic overcharging in some cases.
>
> > # 5. ICML relevance
>
> We respectfully disagree that this is only about financial transparency. The broader issue is **verifiable auditing for ML system services under hidden computation**. Finance is only one visible consequence.
>
> We believe this is relevant to ICML because it concerns:
> - deployment and serving of frontier LLM systems;
> - new auditability problems created by reasoning and agentic APIs.
>
> That said, we agree the paper should explain this connection more clearly, and we will strengthen that framing.
>
> > # 6. Why is auditing information needed at all?
>
> Our point is not that providers force users. Rather, **when one side controls key information and the other side pays but cannot verify it, a standard verification gap appears**.
>
> Our threat model also does **not** assume providers are always malicious. It highlights that **profit incentives plus structural opacity can create systematic misalignment risks**. We will clarify this framing.
>
> > # 7. Missing related work
>
> We agree. Although we cite relevant work (e.g., CoIn, PALACE, model substitution auditing, watermarking, TEE-based ideas), **a dedicated Related Work section is missing**. We will add one in the revision.
>
> > # 8. Relation to low/medium/high reasoning
>
> We agree this needs clarification. These settings are **not** our main object of study; they are only coarse user-visible controls. Even if such levels are exposed, users still cannot know **how many hidden reasoning tokens were actually used**. We will make this distinction clearer.
>
> > # 9. Why focus on tokens/calls rather than parameters/layers?
>
> We are **not** claiming tokens/calls are the only important transparency issue. Our point is that they are the most direct hidden quantities affecting billing and execution in current reasoning/agentic APIs.
>
> Compared with parameter count or architecture details, hidden tokens/calls are:
> - directly tied to the bill and execution path;
> - the clearest case where users may pay for operations they cannot observe.
>
> > # 10. Why is this the right question?
>
> We do **not** claim this is the only important transparency question. Rather, we believe it is **an underexplored but practically important one**.
>
> As reasoning and agentic APIs become more common, users are increasingly charged based on hidden internal computation while seeing only final outputs and limited metadata. This creates a basic gap between **what users pay for** and **what they can observe**. We will state this scope more carefully in the revision.

---

> > ### Author Rebuttal · Reviewer_P6bi · 2026-04-03
> >
> > Thank you for your rebuttal. I will update my score to a 5 (Accept).

---

### Official Review · Reviewer_s8Tg · 2026-03-21

**Significance:** 3
**Argument Clarity:** 3
**Rating:** 5
**Confidence:** 4

**Questions:**

I am confused about the difference claimed by reasoning and agentic APIs. What if the agentic API calls a reasoning model?

**Alternative Views Section:**

Yes

**Compliance With Llm Reviewing Policy A Conservative:**

Affirmed.

**Discussion Potential:**

3

**Final Justification:**

I am in favor of accepting this paper because it clearly lays out the different ways in which LLM platforms can unfairly charge customers and I believe it is very important to start a discussion on addressing the information asymmetry between LLM platforms and customers. The severe opacity of the system allows the platforms to arbitrarily set prices and does not allow customers any visibility into the models/agents that are actually serving them. While the paper does not adequately discuss how market forces could correct the issue, I believe that due to the heavy information asymmetry and high barrier to entry for competitors, it will be very difficult for market forces to actually change anything.

**Paper Summary:**

The paper posits that Commercial Opaque LLM Services (COLS) need to be audited because there is a lot of opportunity for service providers to abuse the opacity by inflating the quantity of tokens/tool calls and/or reducing the quality of responses while charging users for a lower quantity/higher quality. The authors describe how reasoning LLMs can inflate the number of tokens in the reasoning step and how agentic LLMs can inflate then number of tool calls, and propose approaches like commitment-based auditing, predictive auditing and watermarking to address these issues. They then describe how quality can be downgraded by reasoning LLMs using a weaker model, and by agentic LLMs downgrading tools or returning simulated tool responses instead of actually calling the tools, and propose solutions like behavior auditing, signature auditing, and TEE based auditing for this case. The paper concludes with a blueprint for auditing frameworks, and a summary of alternative viewpoints which includes the self-correction in markets, and the discrepancy between the usability and abstraction of COLS on one hand, and the auditability on the other.

**Position:**

Yes

**Position In Title:**

Yes

**Related Work:**

3

**Strengths And Weaknesses:**

Strengths:

The paper provides a solid and plausible summary of the different ways in which COLS can exploit the opacity to charge users at a different service point than the one they actually provide and I think it is very important to expand awareness about this in the AI era when most AI users blindly trust the pricing offered by COLS.

Weakness:

The authors do not really address the alternate viewpoint that the market can self-correct by eliminating providers who overcharge for their services. I believe this is an important issue since there are a lot of opaque industries. A good example is automobile manufacturing where the car companies generally do not disclose their techniques but charge new customers an arbitrary amount. Most people don't really know much about automobile manufacturing and are not even that qualified to evaluate if the price charged by the car maker is justified by its performance. Another example is streaming platforms like Netflix that charge a fixed subscription amount regardless of the quantity/quality of content being watched and we have no way of verifying how much it actually costs them to procure, store or maintain the content, If we can rely on the market to weed out car companies and streaming platforms that are overcharging customers, then why can't we rely on it to weed out the COLS that are inflating quantity/quality?

**Support:**

3

---

> ### Author Rebuttal · Authors · 2026-03-31
>
> > # 1. On whether the market can self-correct without auditing
>
> Thank you for raising this important question. We agree that whether the market can punish dishonest providers is a meaningful alternative view. However, we would like to stress that this is different from the question our paper focuses on, namely **whether the risk itself objectively exists**.
>
> First, our main claim is that in COLS, hidden operations create **real risks of quantity inflation and quality downgrade**. Whether market competition may eventually correct such behavior is a separate question about external discipline, and **does not negate the existence of the risk itself**. In other words, even if the market may remove some dishonest providers in the long run, this does not mean these manipulations are not real technical and economic risks. For example, an API provider may stay ahead because it has a stronger model, while also gaining extra profit through **token inflation, inflated hidden calls, or internal downgrades**. Similarly, when one provider’s model is much stronger than others, its cheaper deployment version (for example, a more compressed or quantized version) may still outperform competitors’ standard services. In this case, market leadership does not automatically prevent internal downgrade. This is exactly why **market advantage and hidden-operation auditing risk are logically independent**.
>
> Second, we also appreciate the reviewer’s examples of automobile manufacturing and streaming platforms. These examples are helpful, but they also highlight why **LLM-as-a-service is different from many traditional opaque industries**. In COLS, many cost-related elements can in fact be measured at a very fine level, such as **token counts, internal model calls, tool calls, and model tiers**. The issue is that these measurable quantities are **systematically hidden from the user**. So the problem is not that the cost is naturally hard to measure, but that **it can be measured precisely while only the provider can see it**. In this sense, the situation is closer to a car manufacturer claiming that an engine the user can never open is built with very expensive materials and therefore costs twice as much, while the user has no way to verify that claim. In our view, the key issue in COLS is that **billing depends directly on intermediate computation that is invisible, unverifiable, and non-disputable to the user**. This is why COLS need **additional auditing mechanisms** more than many traditional industries.
>
> In the revision, we will make this distinction clearer:
> (1) our paper focuses on **whether hidden operations create manipulable risks**;
> (2) whether and to what extent the market can correct such risks is a **separate question** worth discussing, but it does not invalidate the first point.
>
> Our position is therefore **not to deny the role of the market**, but to argue that **once such risks exist and users lack verifiability, market self-correction alone is not a sufficient substitute for auditing**.
>
> > # 2. On the distinction between reasoning APIs and agentic APIs
>
> Thank you also for pointing out the distinction between reasoning APIs and agentic APIs. We agree that this part can be clarified further. Our intent is **not to treat reasoning APIs and agentic APIs as two mutually exclusive categories**. Rather, we view them as **two common forms of COLS**, with different dominant hidden operations and therefore different auditing focuses.
>
> A **reasoning API** mainly involves hidden reasoning within a model, such as long reasoning chains, self-reflection, or branch exploration. Therefore, the main auditing concerns are whether **hidden token counts are inflated** and whether the **quality of the underlying reasoning model is downgraded**.
>
> An **agentic API** mainly involves hidden orchestration across multiple components or agents, such as multiple model calls, agent-to-agent communication, planning loops, and tool use. Therefore, the main auditing concerns are whether **the number of calls is inflated**, whether **tools are actually used**, and whether the **overall workflow is downgraded**.
>
> If an agentic API internally calls a reasoning model, then it naturally contains **both forms of opacity**. In that case, it can be understood as an **agentic COLS with internal reasoning components**, and its audit surface becomes **layered**: one needs to audit both the **hidden reasoning tokens** produced by the reasoning model and the **system-level calls, communication patterns, and tool usage** of the agentic workflow. In other words, the two concepts are **analytically distinct, but they can absolutely coexist in practice**. We appreciate this question and will clarify this point in the revision so that readers do not interpret the two categories as strictly disjoint.

---

> > ### Author Rebuttal · Reviewer_s8Tg · 2026-04-04
> >
> > Thank you for addressing my concerns. The responses are adequate and since I had already recommended accepting the paper, I am keeping my score unchanged.

---

### Official Review · Reviewer_EyLL · 2026-03-22

**Significance:** 3
**Argument Clarity:** 2
**Rating:** 3
**Confidence:** 3

**Questions:**

As in weaknesses.

**Alternative Views Section:**

Yes

**Compliance With Llm Reviewing Policy A Conservative:**

Affirmed.

**Discussion Potential:**

2

**Final Justification:**

Although auditing is challenging as the reviewer has mentioned, the paper has a good position on calling auditing the cost of LLM API calls. I will not change my rating but I am good if the paper gets accepted with improved quality.

**Paper Summary:**

This position paper highlights emerging accountability challenges in Commercial Opaque LLM Services: users are billed for operations they cannot observe, verify, or contest. We formalize two key risks: quantity inflation, where token and call counts may be artificially inflated, and quality downgrade, where providers might quietly substitute lower-cost models or tools. It proposes a modular three-layer auditing framework for COLS and users that enables trustworthy verification across execution, secure logging, and user-facing auditability without exposing proprietary internals.

**Position:**

Yes

**Position In Title:**

Yes

**Related Work:**

2

**Strengths And Weaknesses:**

Strengths:

1. Audit Hidden Operations in Opaque LLM Services is an emerging topic.

2. The proposed analysis methods are helpful to audit the token usage.

3. Three-layer architecture for auditing framework is helpful.

Weaknesses:

1. Reasoning token length prediction sometimes can be hard especially when using dynamic/adaptive thinking mode.

2. Agentic tasks instead of LLM calls are very hard to audit. Agentic tasks are not just tool use. It could be workflow with multiple LLM calls or multi-agent collaboration. Most time it is subscription based and hard for both pricing and auditing.

3. Alternative views are not sufficiently discussed.

**Support:**

2

---

> ### Author Rebuttal · Authors · 2026-03-31
>
> Thank you for the thoughtful feedback. We appreciate your comments and agree that these are important points for improving the paper. Below we clarify our position and how we will revise the paper.
>
> > # 1. Reasoning token length is hard to predict under dynamic/adaptive thinking
>
> We **fully agree** with this point. In fact, this is one of the main challenges we want to highlight in the paper. Our goal is **not** to claim that hidden reasoning token length can be accurately predicted in general. Rather, our point is that **this difficulty itself shows why auditing hidden operations is an important and still underexplored problem**.
>
> In Section 3.3, we already discuss the **high randomness and variability** of LLM behavior as a core challenge for quantity auditing. Even with the same prompt, the number of hidden reasoning tokens or hidden calls can vary a lot, so it is hard to define a stable expected usage. Table 4 also supports this point: even trained prediction models have limited performance across datasets. We do **not** present these results to argue that predictive auditing is already sufficient; instead, we use them to show that **prediction alone is not enough**.
>
> We will revise the paper to make this message clearer. More specifically, we will emphasize that **predictive auditing is only one component, not a complete solution**. A stronger auditing framework should combine multiple methods, such as **commitment-based auditing, behavioral auditing, signature-based auditing, and possibly TEE-based auditing** in high-stakes settings.
>
> > # 2. Agentic tasks are harder to audit than simple LLM calls
>
> We agree with this point as well. Figure 1 already shows that an agentic workflow can itself call a stronger reasoning model, which reflects the **nested structure** of modern agentic systems. More broadly, one important motivation of our paper is that as LLM services move from single-step generation to **agentic workflows**, the hidden execution process becomes much more complex and much harder for users to observe or verify.
>
> We do **not** intend to reduce agentic task auditing to only tool auditing or call auditing. Instead, we agree with the reviewer that **agentic auditing is fundamentally a workflow-level auditing problem**, not only an API-call-level problem. This includes hidden planning, coordination, communication, tool use, and nested model calls.
>
> We also appreciate the reviewer’s point about **subscription-based / credit-based pricing**. As noted around Table 3, many agentic services use coarse-grained billing, which makes it difficult for users to judge the true cost of a single task. We will make this point more explicit in the revision and clarify that auditing in such settings is also about the fairness and consistency of workflow-level resource usage, not only exact per-call accounting.
>
> > # 3. Alternative Views are not discussed enough
>
> We thank the reviewer for pointing this out, and we **agree** that this section is currently too short. In the current draft, we already mention two alternative views:
> (1) market competition and provider reputation may reduce the risk of systematic abuse, so a heavy technical auditing framework may not always be necessary;
> (2) stronger auditing mechanisms may add latency, cost, and system complexity, which could hurt the user experience of modern AI services.
>
> We agree that this discussion should be expanded. In the revision, we plan to add several additional perspectives, including:
>
> - per-query or subscription billing may be a better fit in some markets;
> - some users may accept opacity as a trade-off for convenience and integration;
> - competition and provider reputation may already limit systematic overcharging in some cases.
>
> We believe a fuller discussion of these views will make the paper more balanced and strengthen it as a position paper.
>
> ---
>
> Thank you again for the constructive feedback. We will revise the paper to make these points clearer and to better define the scope and limits of our proposed auditing framework.

---

> > ### Author Rebuttal · Reviewer_EyLL · 2026-04-08
> >
> > I agree with the revision plan from the authors.

---

### Decision · Program_Chairs · 2026-04-30

**Decision:**

Accept (regular)

**Comment:**

I personally found this paper unnecessary in the sense that it addresses what is likely to be a not-long-term pricing mechanism that will evolve naturally to be more transparent (e.g. charge hourly for replacement white collar labour services, rent dedicated capacity, etc.).  That said, I do not feel compelled or justified in overturning the agreement between the reviewers who found the paper timely and were generally appreciative of the auditing framework proposed.